# Chronic kidney disease among people living with HIV on TDF based regimen: A systematic review and meta-analysis

Taklo Simeneh Yazie[1]*, Wondimeneh Shibabaw Shiferaw[2,3], Asaye Alamneh Gebeyehu[4], Assefa Agegnehu Teshome[5], Zenaw Debasu Addisu[6], Abebe Muche Belete[7,8]

1 Department of Pharmacy, Pharmacology and Toxicology, College of Health Sciences, Debre Tabor University, Debre Tabor, Ethiopia, 2 UQ Centre for Clinical Research, Faculty of Medicine, The University of Queensland, Herston, Australia, 3 Department of Nursing, Asrat Weldeyes Health Science Campus, Debre Berhan University, Debre Berhan, Ethiopia, 4 Department of Social and Public Health, College of Health Science, Debre Tabor University, Debre Tabor, Ethiopia, 5 Department of Biomedical Science, College of Health Sciences, Debre Tabor University, Debre Tabor, Ethiopia, 6 Department of Clinical Pharmacy, College of Health Sciences, Bahir Dar University, Bahir Dar, Ethiopia, 7 Department of Biochemistry, West African Centre for Cell Biology of Infectious Pathogens, Cell and Molecular Biology, University of Ghana, Accra, Ghana, 8 Department of Biomedical Science, Asrat Weldeyes Health Science Campus, Debre Berhan University, Debre Berhan, Ethiopia

* taklosimeneh23@gmail.com

**Data Availability Statement:** All relevant data are available within the paper and its supporting materials.

## Abstract

### Introduction

Chronic kidney disease is a major public health concern among people living with human immunodeficiency virus (PLWHIV) who are taking tenofovir disoproxil fumarate-based regimen. Despite the available evidence showing a high prevalence of CKD in this population, comprehensive pooled estimate of CKD among PLWHIV receiving TDF based regimen across the globe is lacking. Hence, the present systematic review aimed to provide a global pooled prevalence estimate of CKD.

### Method

We conducted a systematic review of literatures published between January 2000 and May 2024. Articles and grey literature were searched from the following databases and search engine: PubMed, EMBASE, Scopus, Web of science, The Cumulative Index to Nursing and Allied Health Literature (CINHAL), and Google Scholar. We included eligible studies that report magnitude of CKD in TDF based regimen. We executed the pooled CKD, subgroup analysis, and funnel plot using random effect model. All statistical analysis including sensitivity analysis were made using Stata 17 software.

### Results

Sixty-nine studies with 88299 participants included in this meta-analysis. The pooled prevalence of CKD was 7% (95% CI:6–8). CD4 count less than 200 copies per milliliter, and being female were associated with CKD.

**Funding:** The author(s) received no specific funding for this work.

**Competing interests:** The authors have declared that no competing interests exist.

**Abbreviations:** CG, Cockcroft-Gault; CKD, Chronic Kidney Disease; CKD-EPI, Chronic Kidney Disease-Epidemiology; eGFR, estimated Glomerular Filtration Rate; HIC, High Income Country; JSE, Japanese Society Equation; LIC, Low Income Country; LMIC, Lower Middle-Income Country; LUMIC, Lower Middle-Income Country; MDRD, Modification of Diet in Chronic Kidney Disease; PLWHIV, People Living With HIV; TDF, Tenofovir Disoproxil Fumarate; WHO, World Health Organization.

## Conclusion

We concluded that the magnitude of CKD across the globe is high in people living with HIV who have received TDF based regimen. Early identification of CKD by considering regular renal function monitoring, and risk factors especially low CD4 count, and female gender at birth are essential.

## Trial registration

The protocol has been prospectively registered with PROSPERO ((CRD42020136813).

## Introduction

Chronic kidney disease (CKD) is a common complication among people living with HIV (PLWHIV) [1, 2]. Several evidence indicate that CKD is associated with the use of Tenofovir Disoproxil Fumarate (TDF) [3–7]. The distribution of CKD among PLWHIV on TDF varies across the globe: it ranges from 1–61.7% [8, 9]. A systematic review conducted by Cooper RD et al. in 2010 [10], and Mtisi TJ et al. in 2019 [11] confirmed the association of TDF utilization with the presence of kidney function loss. However, since then, there has been no recent comprehensive evidence elucidating the burden of CKD among PLWHIV on TDF.

Therefore, this review aims to determine the prevalence of CKD among PLWHIV receiving a TDF based regimen. Thus, the finding of this review will provide valuable insights into the management of people living with HIV patients on TDF and help policy makers and clinician to make informed decisions about the use of this drug in this patient's population.

## Methods

This review was conducted following the guidance and instructions outlined in the Preferred Reporting Items for Systematic Review and Meta-analysis (PRISMA) [12] (S1 Table). The study protocol (CRD42020136813) was prospectively registered with PROSPERO. The chapter on systematic reviews of prevalence and incidence studies in Joanna Briggs Institute (JBI) Reviewer's Manual for prevalence studies, and condition, context, and population (CoCoPop) was used to form the review questions [13]. Hence, this review's research questions are: i) what is the prevalence of CKD among PLWHIV on TDF based regimen? ii) what factors contribute to CKD among PLWHIV on TDF?

### Eligibility criteria

Following CoCoPop framework, the eligibility criteria for the review are described as follows:

**Population:** We included studies involving participants (age 13 years or older) living with HIV and receiving TDF based regimen. These age group patients have comparable renal function [14] and TDF dose use [15]. We excluded studies involving participants with age less than 13 years.

**Condition:** We considered studies that report the main outcome of the study (i.e., prevalence of CKD among PLWHIV on TDF).

**Context:** We included studies conducted in community and institution-based studies that report the prevalence of CKD.

**Types of studies:** We included Clinical trial and observational studies (cross sectional, cohort (retrospective, and prospective). We excluded case series, and case reports. We included studies published from January 1, 2000 to May 2, 2024. We considered only studies published in English.

## Information source

To conduct this review, the search strategy employed was the Peer Review of Electronic Search Strategies (PRESS) methodology for systematic reviews in our search strategy [16]. To undertake the search strategy, the primary investigators initially developed search terms. Subsequently, all co-authors reviewed and approved the comprehensive search terms. The databases used were PubMed, Scopus, EMBASE, Cumulative Index to Nursing and Allied Health Literature (CINHAL), and Web of Science. Additionally, we examined reference lists in papers for relevant papers. Furthermore, we searched papers from grey literature like Google scholar.

**Search strategies.** Initial key words used were HIV, Chronic kidney disease, and TDF. Following this, independent search terms were developed for each key words, for HIV included "HIV" OR "hiv" OR "human immunodeficiency virus" OR"AIDS "OR"acquired immunodeficiency syndrome." Search terms for chronic kidney disease include **"**Chronic Kidney Failure" OR "Chronic Renal Failure" OR "Chronic kidney disease" OR "End-Stage Kidney Disease" OR "End-Stage Renal Disease" OR "End-Stage Renal Failure" OR "ESRD" OR "Renal Insufficiency" OR "Renal impairment" OR "Kidney impairment" OR "Renal failure" OR "Kidney failure" OR "Renal dysfunction" OR "Kidney dysfunction." Search terms for TDF includes "Tenofovir" OR "Tenofovir Disoproxil Fumarate" OR "TDF." The full search term is in included in the (S2 Table).

**Selection of studies.** Initially, the articles found from each database were imported into Endnote version 8.1 citation manager software. Duplicate articles were then removed. Following this, the titles and abstracts of each article were assessed for inclusion by Two (TSY and AMB) independent review authors. Additionally, articles deemed suitable for the full-text review were evaluated for inclusion against the pre-identified inclusion criteria by other two review co-authors (AAG and WSS). Any disagreements arising during the selection process were resolved with consultation of a third review author (ZDA).

**Methodological quality assessment.** The included studies were evaluated methodological quality using the Newcastle-Ottawa Scale (NOS) tool for observational [17], and version 2 of the Cochrane risk-of-bias tool for randomized trials (RoB 2) for randomized controlled trial studies [18]. In each included studies we assessed representativeness, response rate, method, comparability of the subject and the appropriateness of statistically analysis used for observational studies, and randomization, allocation concealment, adherence to intervention, and outcome assessment for clinical trials. Two review authors (WSS and AAT) checked quality of the studies using the above criteria. Any disagreement was resolved through discussion (S3 Table).

**Data extraction.** We used the Joanna Briggs Institute (JBI) extraction form for prevalence and incidence studies available in Munn et al [13]. Two authors (AAG and WSS) performed data extraction. The following study characteristics were extracted from included studies: first author, publication year, region, study design, sample size, and outcome reported (i.e. CKD) and associated factors like age, sex, CD4 count, and glomerular filtration rate. We resolved disagreements by consensus or discussion.

## Missing data handling

In this review, we used a complete-case analysis approach to include only studies that had complete data for the outcome of interest. For studies with missing data, we tried to contact the corresponding authors to obtain the missing information. However, despite these efforts, none of them responded to our requests.

**Approaches to CKD diagnosis.** There is no internationally standardized eGFR estimation method recommended to be used universally across the world. Hence, studies used different eGFR estimation method to assess renal function. These are Cockcroft-Gault (CG) [19], Modification of Diet in Renal Diseae (MDRD) [20], Chronic Kidney Disease Epidemiology (CKD-EPI) [21], Modification of Diet in Renal Disease (MDRD) without race factor [22], and Japanese Society (JSE) [23]. The presence of proteinuria or kidney damage confirmed via imaging alone or eGFR<60ml/min persisted for at least 3 months can be used to diagnose CKD [24]. Treatment guidelines recommend eGFR or CrCl <50 [15] or 60ml/min [25, 26] to define CKD, and to modify or avoid TDF use in HIV care, so we included studies that used eGFR or CrCl level of either <50 or 60 ml/min. CrCl/eGFR<50/60ml/min occurred onspot or persisted at least 3 months post TDF initiation was used as CKD diagnostic criterion to loosen the inclusion criteria for the purpose of revealing the pooled estimate of CKD across the globe.

**Assessment of risk of bias.** Each included study was evaluated using Hoy risk of bias assessment tool for reporting prevalence data [27]. The Hoy score is marked out of ten and a value of 8–10 indicated low bias, 5–7 moderate bias and ≤4 high bias. Two review authors (AAG and TSY) independently assessed the risk of bias.

**Heterogeneity and publication bias.** We used Cochran's $Q$ and $I^2$ statistics to measure heterogeneity among the studies included in each analysis [28]. Higgins et al. suggest that an $I^2$ value of 25%, 50%, and 75% indicate low, medium, and high heterogeneity, with respective order. We performed subgroup analysis based on region (continent), CKD diagnostic criteria, income level, and study design. We also performed sensitivity analysis was also conducted for each study's effect on the overall prevalence. Funnel plot was used to visually inspect publication bias. Egger's test was used to assess statistical significance of publication bias [29].

**Statistical analysis.** DerSimonian–Laird random-effects models [30] was used to generate the pooled prevalence of CKD. The pooled effect size (i.e., prevalence) with weighted and their 95% confidence interval (CI) was generated. Additionally, the pooled effect size (i.e., odds ratio for age, and eGFR; hazard ratio for sex, and CD4 count) also generated with their 95% CI. We displayed all analysis in the form of forest plot. We used Stata software version 17.

# Results

## Description of included studies

**Search results.** We access 1493 studies from databases and manual search. After removal of duplicates, 1256 studies remained, and we excluded 1027 studies during title and abstract screening stage. We reviewed the remaining 229 studies for full-text eligibility, and excluded 160 due to various reasons (Fig 1).

## Characteristics of included studies

We included sixty-nine studies comprising of 88299 study participants in the analysis. Among them, three clinical trial [31–33], fifteen prospective cohort [20–22, 34–45], thirty-five retrospective cohort [6, 8, 9, 19, 23, 46–75], and sixteen cross sectional [76–91]. The sample size in the included studies ranges from 38 [61] to 11153 [45]. According to the World Bank geographical classification [92], we found twenty-four studies in East Asia & Pacific, twenty-one

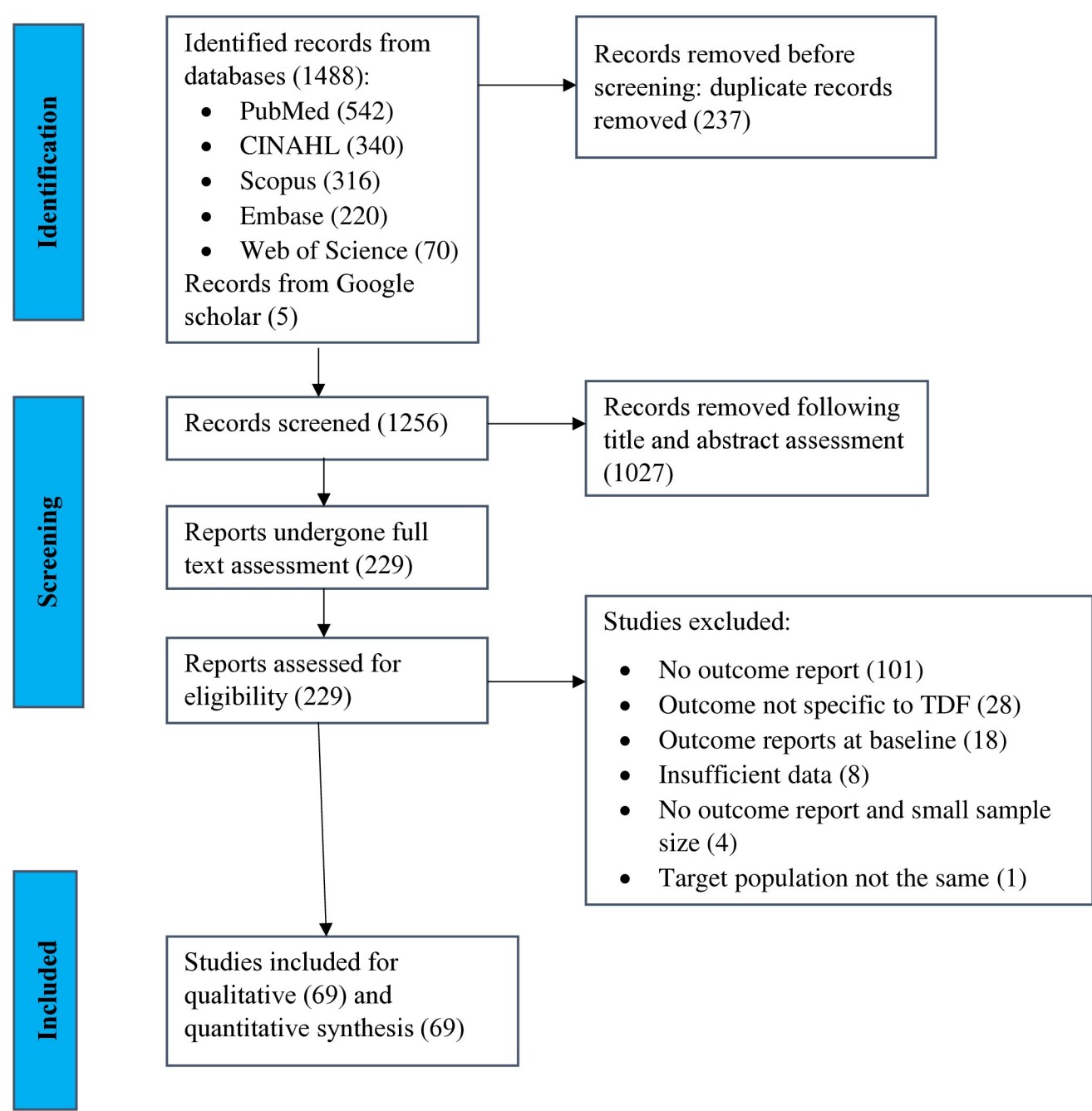

**Fig 1. PRISMA flow diagram describing the selection processes of eligible studies.**

in Sub-Saharan Africa, thirteen in Europe & Central Asia, eight in North America, and three in more than one region. We reviewed studies conducted: seven each from Japan, and US; four each from Ethiopia, China, Ghana, Italy, Spain, and Thailand; three each from Australia, and France; two each from South Africa, Cameroon, Nigeria, South Korea, United Kingdom, and Namibia; and one study each in Malawi, Eastern and southern African countries, Uganda, Malaysia, India-UK, Myanmar, Zambia, Singapore, Canada and New Zealand, Tanzania,

Asian countries, and Vietnam. The included study characteristics including study region, sample size, and eGFR estimation methods are presented in Table 1 below.

## Prevalence of chronic kidney disease

In the present meta-analysis, we used sixty-nine studies to estimate the pooled prevalence of CKD. Based on Hoy D et al., we found three (4.4%) studies with moderate risk of bias, and sixty-six (95.5%) studies with low risk of bias. The overall pooled prevalence of CKD diagnosed with estimated glomerular filtration rate (eGFR) was 7% (95% CI: 6%-8%), $I^2$ = 98.54%, p < 0.01 (Fig 2). This I squared result showed high heterogeneity among studies, which indicates the necessity of subgroup analysis.

## The prevalence of CKD by its diagnostic crteria

All but two included studies used only estimated glomerular filtration rate (eGFR) or creatine clearance (CrCl)<50/60ml/min to diagnose CKD. Two studies used proteinuria in addition to eGFR or CrCl<50/60ml/min to estimate CKD [44, 88].

In this meta-analysis, according to the eGFR cutoff point of <50 versus <60ml/min/average adult body surface area, the pooled estimate of CKD was 6% (95% CI: 4–7%), and 7% (95% CI: 6–8%), respectively. Heterogeneity between studies in both groups were found to be high; however, regarding the pooled CKD etstimate, there was no significant difference (P = 0.19) between groups.

In the present review concerning eGFR or CrCl <50/60ml/min persistence to define CKD, thirty five studies used on the spot estimation (eight with no baseline eGFR data and twenty seven with normal baseline eGFR), while thirty four studies used a two time point estimation at least 3 months apart. According to these diagnostic criteria, the pooled prevalence of CKD was 9% (95% CI: 6–12%) diagnosed with on the spot eGFR with no baseline data, 9% (95% CI: 7–11%) confirmed with on the spot eGFR with normal baseline data, and 5% (95% CI: 4–6%) in cases with <50 or 60ml/min confirmed at least 3 months apart. We found high heterogeneity and significant difference between studies in all subgroups and between groups.

Regarding eGFR estimation equations, among the included studies, 27 used CKD-EPI, 20 used MDRD, 13 used CG, 5 used MDRD without race factor, and 4 used JSE equation to estimate eGFR. In subgroup analysis of the pooled prevalence of CKD based on eGFR estimation equations, we found a prevalence of 13% (95% CI: 8–18%) in studies that used CG, 10% (95% CI: 5–15%) in studies that used JSE, 7% (95% CI: 5–8%) in those using CKD-EPI, 4% (95% CI: 3–6%) with MDRD, and 3% (95% CI: 1–5%) with MDRD without race factor. In these subgroup analyses, we found high heterogeneity between studies in all groups, with significant group differences. Additionally, we found that pooled prevalence of CKD based on CG, MDRD, and MDRD without race factor was significantly different from the overall pooled prevalence estimate of CKD (Table 2).

**CKD in PLWHIV by study design, age group, region and income.** The pooled prevalence of CKD based on study design was as follows: 9% (95% CI: 6–11%) in cross sectional studies, 7% (95% CI: 6–9%) in retrospective cohort studies, 5% (95% CI: 4–6%) in prospective cohort studies, and 3% (95% CI: 2–4%) in clinical trials. We found high heterogeneity accross all study groups except clinical trials. Pooled prevalence of CKD in prospective cohort studies, and clinical trials were significantly different from the overall pooled prevalence. We found significant difference between the study design groups.

Among the included studies, six were conducted in low-income countries (LIC), thirteen in lower middle-income countries (LMIC), twelve in upper middle-income countries (UMIC), thirty-five in high income countries (HIC), and three in others (from LIC (n = 1), LMIC,

**Table 1. Characteristics of included studies.**

| Author-year | Continent | Study design | Sample size | eGFR equation | eGFR cutoff point |
|---|---|---|---|---|---|
| Cournil A et al. 2017 [32] | Sub-Saharan Africa | RCT | 275 | MDRD | <60ml/min |
| Chikwapulo B et al., 2018 [19] | Sub-Saharan Africa | RetroCT | 426 | CG | <50ml/min |
| Yazie TS et al., 2019 [35] | Sub-Saharan Africa | ProCohort | 63 | CKD-EPI | <60 ml/min |
| Mwafongo A et al., 2015 [31] | Sub-Saharan Africa | RCT | 741 | CG | <50 ml/min |
| Zachor H et al., 2016 [50] | Sub-Saharan Africa | RetroCT | 650 | CKD-EPI | <60 ml/min |
| Ojen BV et al., 2018 [49] | Sub-Saharan Africa | RetroCT | 3214 | MDRD | <60 ml/min |
| Nartey ET et al., 2019 [51] | Sub-Saharan Africa | RetroCT | 300 | CG | <50 ml/min |
| Nyende L et al., 2020 [79] | Sub-Saharan Africa | CS | 278 | CKD-EPI | < 60ml/min |
| Neary M et al., 2020 [34] | Sub-Saharan Africa | ProCohort | 66 | CG | <60mL/min |
| Bock P et al., 2019 [47] | Sub-Saharan Africa | RetroCT | 1634 | MDRD worf | < 60 mL/min |
| Belete AM et al., 2021 [77] | Sub-Saharan Africa | CS | 243 | CKD-EPI | <60 ml/min |
| Debeb SG et al., 2021 [48] | Sub-Saharan Africa | RetroCT | 200 | CKD-EPI | <60 ml/min |
| Fritzsche C et al.,2017 [76] | Sub-Saharan Africa | CS | 119 | CKD EPI | < 60/mL/min |
| Chadwick D et al., 2015 [78] | Sub-Saharan Africa | CS | 101 | CG | <60 ml/min |
| Kim JH et al., 2022 [39] | East Asia & Pacific | ProCohort | 392 | MDRD | < 60/mL/min |
| Nishijima T et al., 2016 [38] | East Asia & Pacific | ProCohort | 417 | CKD-EPI | <60 ml/min |
| Young et al., 2007 [40] | North America | ProCohort | 593 | CG | <50 ml/min |
| Feng L et al., 2022 [20] | East Asia & Pacific | ProCohort | 622 | MDRD | <60 ml/min |
| Sutton SS et al., 2020 [60] | North America | RetroCT | 4475 | CKD-EPI | <60 ml/min |
| Cheung J et al., 2018 [21] | East Asia & Pacific | ProCohort | 985 | CKD-EPI | <60 ml/min |
| Tan LKK et al., 2009 [61] | Europe & Central Asia | RetroCT | 38 | MDRD | <60 ml/min |
| Milazzo L et al., 2016 [62] | Europe & Central Asia | RetroCT | 78 | CKD-EPI | <60 ml/min |
| Kalemeera F et al., 2020 [52] | Sub-Saharan Africa | RetroCT | 6744 | CKD-EPI | <50 ml/min |
| Calza L et al., 2014 [84] | Europe & Central Asia | CS | 409 | MDRD | <60 ml/min |
| Quesada PR et al., 2015 [41] | Europe & Central Asia | ProCohort | 451 | MDRD | <60 ml/min |
| Low JZ et al., 2018 [63] | East Asia & Pacific | RetroCT | 314 | MDRD | <60 ml/min |
| Pujari SN., 2014 [53] | South Asia-Europe & Central Asia | RetroCT | 1225 | MDRD | <60 ml/min |
| Jotwani V et al., 2016 [81] | North America | CS | 573 | CKD-EPI | <60 ml/min |
| Visuthrankul J et al., 2021 [54] | East Asia & Pacific | RetroCT | 700 | MDRD | <60 ml/min |
| Nishijima T et al., 2014 [36] | East Asia & Pacific | ProCohort | 422 | JSN equation | <60 ml/min |
| Kyaw NTT et al., 2015 [56] | East Asia & Pacific | RetroCT | 1372 | CG | <50 ml/min |
| O'Donnel EP et al., 2011 [55] | North America | RetroCT | 348 | MDRD | <60 ml/min |
| Nishijima T et al., 2011 [57] | East Asia & Pacific | RetroCT | 495 | MDRD | <60 ml/min |
| Okpa HO et al., 2019 [80] | Sub-Saharan Africa | CS | 60 | CG | <60 ml/min |
| Woolnough EL et al., 2018 [58] | East Asia & Pacific | RetroCT | 473 | CKD-EPI | <60 ml/min |
| Nishijima T et al., 2017 [82] | East Asia & Pacific | CS | 774 | JSN equation | <60 ml/min |
| Obiri-Yeboah D et al., 2018 [83] | Sub-Saharan Africa | CS | 288 | MDRD | <60 ml/min |

*(Continued)*

**Table 1.** (Continued)

| Author-year | Continent | Study design | Sample size | eGFR equation | eGFR cutoff point |
|---|---|---|---|---|---|
| Lapadula G et al., 2016 [37] | Europe & Central Asia | ProCohort | 2023 | CKD-EPI | <60 ml/min |
| Hsu R et al., 2020 [59] | North America | RetroCT | 6222 | CKD-EPI | <60 ml/min |
| Morlat P et al., 2013 [22] | Europe & Central Asia | ProCohort | 3268 | MDRD worf | <60 ml/min |
| Chabala FW et al., 2021 [42] | Sub-Saharan Africa | ProCohort | 201 | CKD-EPI | <60 ml/min |
| Likanonsakul S et al.,2016 [85] | East Asia & Pacific | CS | 273 | CKD-EPI | <60 ml/min |
| Flandre P et al.,2016 [43] | Europe & Central Asia | ProCohort | 3543 | MDRD worf | <60 ml/min |
| Lee KH et al.,2017 [65] | East Asia & Pacific | RetroCT | 50 | MDRD worf | <60 ml/min |
| Paengsai N et al.,2022 [66] | East Asia & Pacific | RetroCT | 8710 | CKD-EPI | <60 ml/min |
| Suzuki S et al.,2017 [6] | East Asia & Pacific | RetroCT | 720 | CKD-EPI | <60 ml/min |
| Campbell LJ et al.,2009 [46] | Europe & Central Asia | RetroCT | 843 | MDRD | <60 ml/min |
| Domingo P et al.,2019 [64] | Europe & Central Asia | RetroCT | 4852 | CKD-EPI | <60 ml/min |
| Ando M et al., 2011 [44] | East Asia & Pacific | ProCohort | 244 | JSN equation | <60 ml/min |
| Chua AC et al., 2012 [67] | East Asia & Pacific | RetroCT | 154 | CG | <50 ml/min |
| Nishijima T et al., 2015 [23] | East Asia & Pacific | RetroCT | 703 | JSN equation | <60 ml/min |
| Ahmed E et al., 2020 [86] | Sub-Saharan Africa | CS | 290 | CG | <60 ml/min |
| Chan A et al., 2019 [33] | North America-East Asia & Pacific | OLCT | 335 | CG | <50 ml/min |
| Huang Y et al., 2017 [8] | East Asia & Pacific | RetroCT | 391 | CKD-EPI | <60 ml/min |
| Juega-Mariño J et al., 2017 [87] | Europe & Central Asia | CS | 699 | MDRD | <60 ml/min |
| Mwemezi O et al., 2020 [88] | Sub-Saharan Africa | CS | 249 | CKD-EPI | <60 ml/min |
| Reynes J et al., 2013 [89] | Europe & Central Asia | CS | 658 | MDRD | <60 ml/min |
| Monteagudo-Chu et al., 2012 [69] | North America | RetroCT | 111 | MDRD | <60 ml/min |
| Medland NA et al., 2017 [71] | East Asia & Pacific | RetroCT | 1442 | CKD-EPI | <50 ml/min |
| Suppadungsuk S et al., 2022 [9] | East Asia & Pacific | RetroCT | 141 | CKD-EPI | <60 ml/min |
| Calza L et al.,2013 [68] | Europe & Central Asia | RetroCT | 235 | MDRD | <60 ml/min |
| Pedrol E et al., 2015 [72] | Europe & Central Asia | RetroCT | 73 | CKD-EPI | <60 ml/min |
| Kalemeera F et al., 2023 [73] | Sub-Saharan Africa | RetroCT | 7526 | CG | <60 ml/min |
| Joshi et al., 2019 [74] | South Asia-East Asia & Pacific | RetroCT | 703 | CKD-EPI | <60 ml/min |
| Hoang C et al., 2020 [90] | East Asia & Pacific | CS | 400 | CKD-EPI | <60 ml/min |
| Mocroft A et al., 2015 [45] | Middle East &North America, Latin America & Caribbean, North America, East Asia & Pacific, Europe & Central Asia | ProCohort | 11153 | CG | <60 ml/min |
| Crum-Cianflone N et al., 2010 [91] | North America | CS | 318 | MDRD | <60 ml/min |
| Liu F et al., 2021 [75] | East Asia & Pacific | RetroCT | 797 | MDRD worf | <60 ml/min |
| Yang J et al., 2019 [70] | East Asia & Pacific | RetroCT | 414 | MDRD | <60 ml/min |

**Abbreviations:** CS: Cross sectional; ProCohort: Prospective cohort; RetroCT: Retrospective cohort; RCT: Randomized controlled trial; OLCT: Open label clinical trial; MDRD: Modification of diet in renal disease equation; CKD-EPI: Chronic kidney disease epidemiology collaboration equation; MDRD worf: MDRD equation without race factor; CG: Cockcroft-Gault; JSN equation: Japan society of nephrology equation

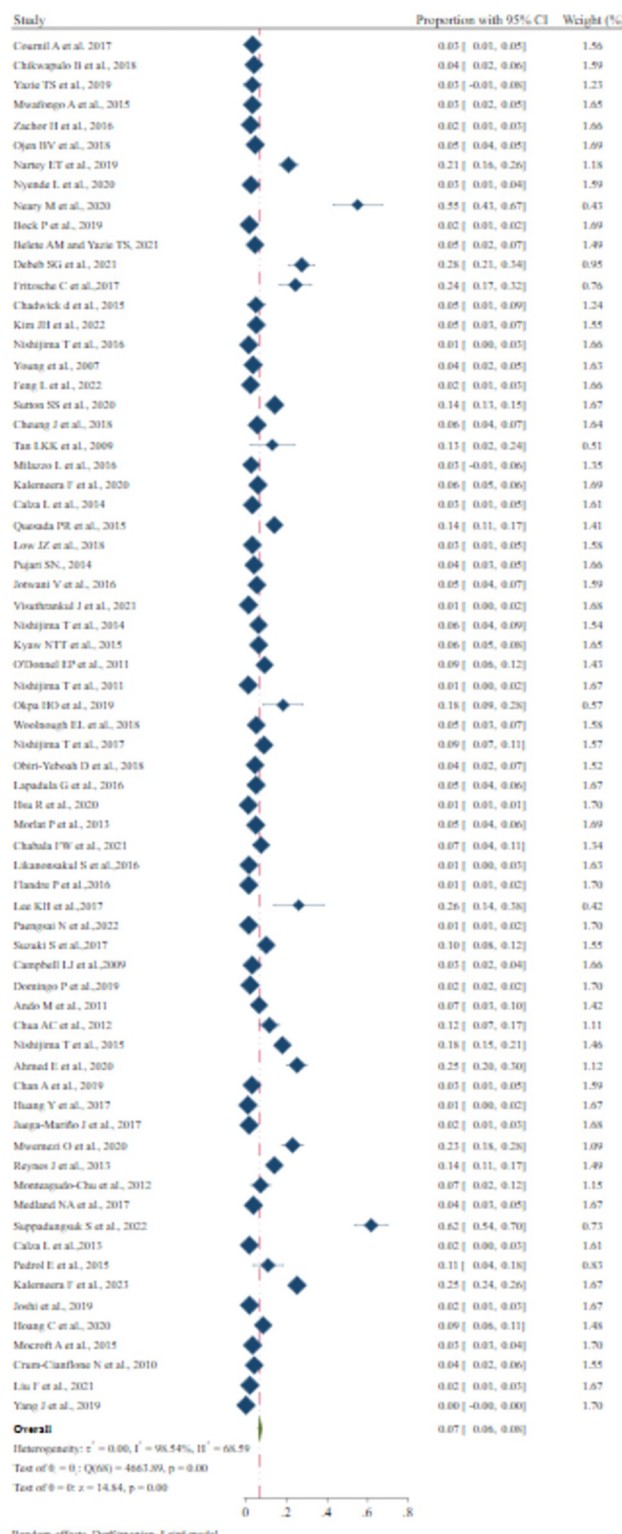

**Fig 2. Overall pooled proportion of included studies.**

**Table 2. CKD based on CKD diagnostic approach.**

| Variable | Category of variable | Pooled estimate of CKD (95% CI) | tau2 | % I2 | H2 | df | Q | P | Test of group differences |
|---|---|---|---|---|---|---|---|---|---|
| eGFR cutoff point | <50ml/min | 0.057(0.042–0.073) | 0.000 | 91.13 | 11.28 | 8 | 90.20 | 0.000 | Chi2(1) = 1.74, P = 0.187 |
| | <60ml/min | 0.070(0.060–0.080) | 0.001 | 98.65 | 74.06 | 59 | 4369.81 | 0.000 | |
| eGFR<50/60ml/min confirmation | On the spot[a] | 0.088(0.055–0.120) | 0.002 | 93.01 | 14.32 | 7 | 100.21 | 0.000 | Chi2(2) = 12.23, P = = 0.002 |
| | On the spot[b] | 0.088(0.067–0.109) | 0.003 | 99.21 | 126.40 | 26 | 3286.40 | 0.000 | |
| | =/>3month | 0.053(0.043–0.062) | 0.001 | 97.36 | 37.84 | 33 | 1248.72 | 0.000 | |
| eGFR estimation | CG | 0.131(0.079–0.183) | 0.009 | 99.38 | 160.80 | 12 | 1929.62 | 0.000 | Chi2(4) = 26.71, P = 0.000 |
| | MDRD | 0.044(0.03–0.056) | 0.001 | 95.86 | 24.13 | 19 | 458.51 | 0.000 | |
| | CKD-EPI | 0.066(0.054–0.079) | 0.001 | 98.05 | 51.21 | 26 | 1331.46 | 0.000 | |
| | MDRD worf | 0.029(0.012–0.045) | 0.000 | 95.26 | 21.08 | 4 | 84.32 | 0.000 | |
| | JSE | 0.099(0.050–0.148) | 0.002 | 93.52 | 15.42 | 3 | 46.26 | 0.000 | |

**Abbreviation**: MDRD: Modification of diet in renal disease equation; CKD-EPI: Chronic kidney disease epidemiology collaboration equation; MDRD worf: MDRD equation without race factor; CG: Cockcroft-Gault; JSN equation: Japan society of nephrology equation.

[a]On the spot CKD confirmation without baseline eGFR,

[b]On the spot CKD confirmation with normal baseline eGFR.

UMIC, and HIC (n = 2)). These studies showed high heterogeneity, with significant difference between income groups. Studies from LMIC, and those spanning multiple income categories showed a significant difference in pooled CKD prevalence compared to the overall pooled prevalence of CKD.

We included twelve studies with participants aged thirteen years and above, and fifty-seven studies with participants aged eighteen years and older. Subgroup analysis did not show significant difference in pooled prevalence of CKD between these age groups. We found 9% (95% CI: 6–13%), and 6% (95% CI: 5–7%) pooled prevalence of CKD in age groups of thirteen years and above, and eighteen years and older group, respectively.

The highest pooled prevalence of CKD was found in Sub-Sahara Africa (11.7% [95% CI: 8.4–15%]), while the lowest was in the others regions group (3% [95% CI: 2–4.1%]). We found significantly different pooled prevalence of CKD in Sub-Sahara Africa, and others regions group compared to the overall pooled prevalence of CKD. We found high heterogeneity between studies (P<0.05)) from all group of regions. We also found significant difference between region groups (Table 3).

**Meta-analysis of factors associated with CKD.** In our meta-analysis, we used six studies to determine the pooled effect of predictor variables. We reported the pooled odds effect for age >50 years (OR = 1.13, 95% CI: 0.05–26.00) [46, 77], and eGFR in the range of 60-79ml/min (OR = 6.04, 95% CI: 0.97–37.72) [46, 47], each based on two studies. These two factors did not show a significant association with CKD (Figs 3 and 4).

In the present meta-analysis, we found a pooled hazard ratio of CD4 count less than 200 (HR = 2.54, 95% CI: 1.41–4.58) compared with higher CD4 counts in two studies [50, 56], and of being female (HR = 1.91, 95% CI: 1.56–2.35) compared with being male in three studies [50, 52, 56]. These factors were significantly associated with CKD (Figs 5 and 6).

## Description of factors associated with CKD in included studies

Several factors were associated with CKD in the included studies; however, we did not present their pooled effect because the studies used different category of independent factors, different statistical analysis methods or both.

**Table 3. CKD in PLWHIV by study design, age group, region and income.**

| Variable | Category of variable | Pooled estimate of CKD (95% CI) | tau$^2$ | % I$^2$ | H$^2$ | df | Q | P | Test of group differences |
|---|---|---|---|---|---|---|---|---|---|
| Study design | Cross sectional | 0.086 (0.061–0.110) | 0.002 | 94.73 | 18.97 | 15 | 284.57 | 0.000 | Chi2(3) = 33.13, P = 0.000 |
| | Retro cohort | 0.073(0.059–0.087) | 0.002 | 99.15 | 117.04 | 34 | 3979.38 | 0.000 | |
| | Prospective cohort | 0.051(0.039–0.063) | 0.000 | 94.86 | 19.46 | 14 | 272.48 | 0.000 | |
| | Clinical trial | 0.032(0.023–0.041) | 0.000 | 0.00 | 1.00 | 2 | 0.06 | 0.971 | |
| Income level | Low | 0.106(0.047–0.164) | 0.005 | 95.91 | 24.44 | 5 | 122.21 | 0.000 | Chi2(4) = 47.44, P = 0.000 |
| | Lower middle | 0.109(0.083–0.135) | 0.002 | 94.19 | 17.21 | 12 | 206.55 | 0.000 | |
| | Upper middle | 0.071(0.044–0.098) | 0.002 | 99.62 | 260.99 | 11 | 2870.84 | 0.000 | |
| | High | 0.058(0.047–0.069) | 0.001 | 97.08 | 34.21 | 34 | 1163.09 | 0.000 | |
| | Other | 0.028(0.018–0.038) | 0.000 | 75.17 | 4.03 | 2 | 8.05 | 0.018 | |
| Age | >/ = 13 years | 0.092(0.058–0.125) | 0.003 | 99.62 | 263.54 | 11 | 2898.89 | 0.000 | Chi2(1) = 2.91, P = 0.088 |
| | >/ = 18 years | 0.061(0.053–0.069) | 0.001 | 96.81 | 31.39 | 56 | 1757.89 | 0.000 | |
| Region | Sub-Sahara Africa | 0.117(0.084–0.150) | 0.005 | 99.02 | 101.93 | 20 | 2038.58 | 0.000 | Chi2(2) = 29.81, P = 0.000 |
| | East Asia & Pacific | 0.056(0.045–0.067) | 0.001 | 97.24 | 36.17 | 23 | 832.01 | 0.000 | |
| | Europe & Central Asia | 0.047(0.034–0.060) | 0.000 | 94.98 | 19.93 | 12 | 239.21 | 0.000 | |
| | North America | 0.059(0.016–0.103) | 0.004 | 98.88 | 89.12 | 7 | 623.82 | 0.000 | |
| | Other | 0.030(0.020–0.041) | 0.000 | 80.36 | 5.09 | 2 | 10.18 | 0.006 | |

**Age.** Regarding age, different statistical methods were used to determine the association between age and the presence of CKD. Participants aged 42–53 years, compared to those aged 18–29 years (OR = 3.1, [95% CI: 1.12–8.55]) [86], and those older than 45 years, compared to 18–35 years (OR = 3.25. 95% CI: 1.3–8.14 [47], were three times more likely to experience CKD. Study participants had almost 4% increased risk of developing CKD (RR = 1.05, 95% CI: 1.01–1.09) [19];(RR = 1.04, 95% CI: 1.03–1.06) [51] for every one year increase in age. Participants older than 45 years were three times more likely to have CKD compared to those aged 18–35 years (HR = 3.4, 95% CI: 2.2–5.2) [56], while an increase in age by ten years was

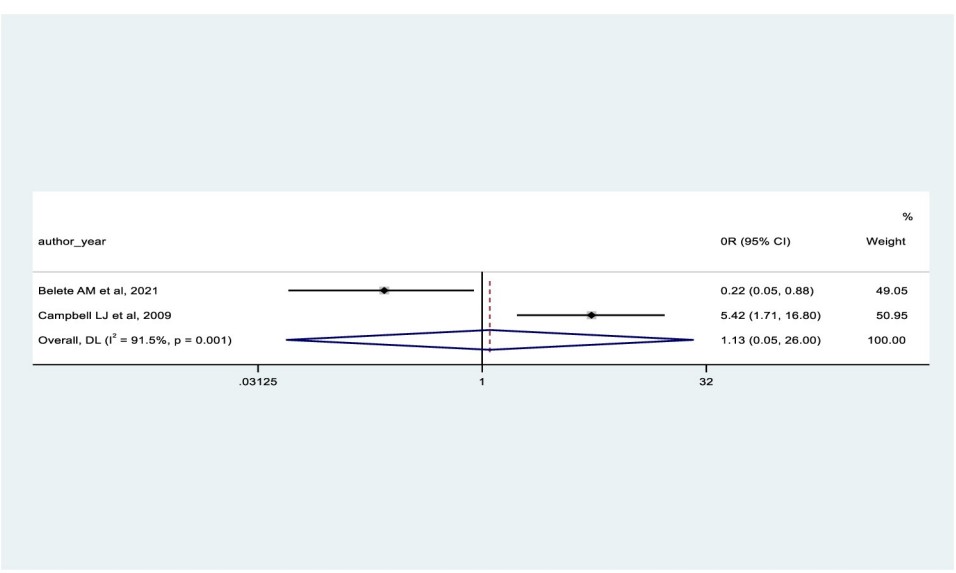

**Fig 3. Shows the forest plot on the association between age of participants with presence of CKD.**

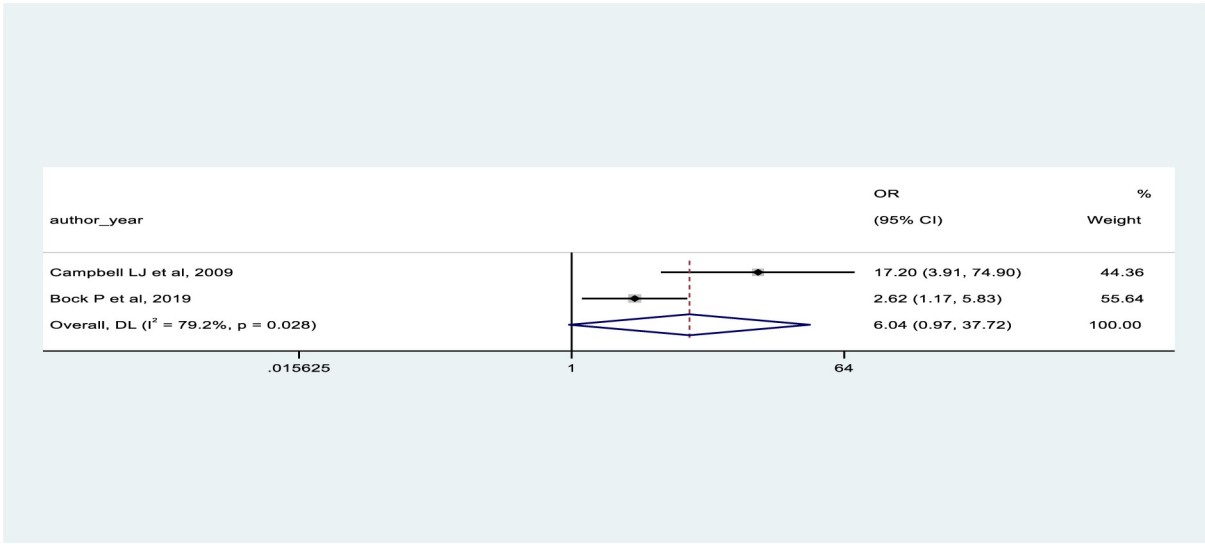

**Fig 4. Shows the forest plot on association between eGFR 60–89 and presence of CKD.**

associated with almost twice the risk of experiencing CKD (HR = 2.21, 95% CI: 1.61–3.05(53); HR = 1.9, 95% CI: 1.1–3.29(50)).

**Body mass index (BMI).** BMI was a significant predictor of CKD in some included studies. Study participants with a lower BMI were positively associated with the presence of CKD compared to their counterparts. Specifically, a BMI of <18.5 kg/m$^2$ compared to >/ = 18.5 kg/m$^2$ (OR = 4.39, 95% CI: 2.24–8.61) [86], <18.5 kg/m$^2$ compared to 18.5–24.9 kg/m$^2$ (RR = 3.87, 95% CI: 2.49–6.03) [51], a BMI of <16 kg/m$^2$ (HR = 2.3, 95% CI: 1.1–5), and 16–18.5 (HR = 1.8, 95% CI: 1.1–3.2) compared to 18.5–24.9 kg/m$^2$ [56] were associated with CKD.

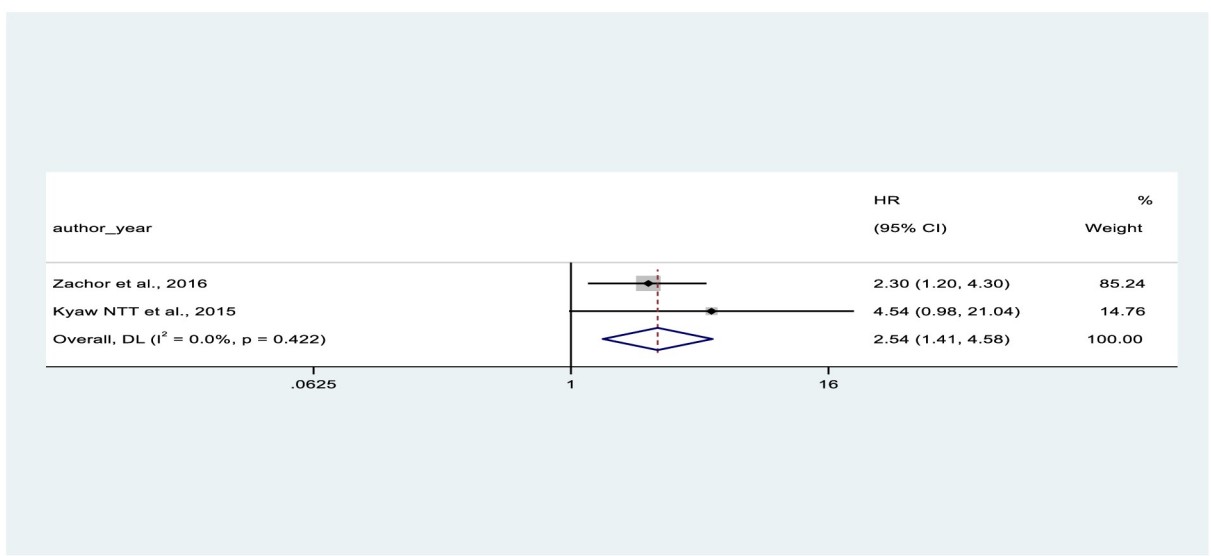

**Fig 5. Shows the forest plot on association between CD4 count of study participants and CKD.**

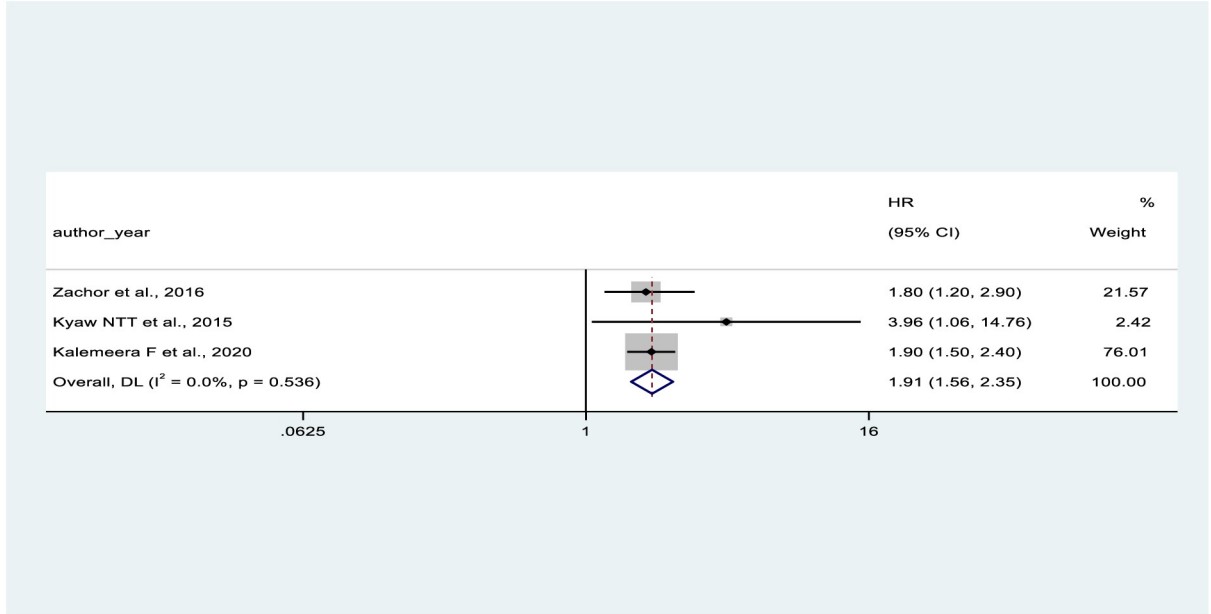

**Fig 6. The forest plot on the association between gender at birth on CKD.**

**World Health Organization (WHO) clinical stage.** Concerning the WHO HIV/AIDS clinical stages, the risk of experiencing CKD was almost three times for individuals in stage III (RR = 3.78, 95% CI: 1.42–10.06), and stage IV (RR = 3.42, 95% CI: 1.16–10.9) compared to those in stage I [51]. On the other hand, another study found a negative association between clinical stage IV and the presence of CKD compared to stage I (OR = 0.1, 95% CI: 0.03–0.36 [86].

**Comorbidity.** The presence of high diastolic blood pressure, cancer, and diabetes mellitus as comorbidities were positive predictors of the presence of CKD compared to their counterparts. specifically, DBP >100 mmHg (RR = 2.78, 95% CI: 1.02–7.58) [19], cancer (OR = 18.2, 95% CI: 122–271.7) [77], and DM (HR = 3.6, 95% CI: 1.6–8.2) [56] were significantly associated with the presence of CKD.

**Antiretroviral drug class, prior ART exposure, and viral load.** Regarding the antiretroviral drug class, participants who have received TDF with ritonavir boosted protease inhibitors (PI/r) had higher risk of developing CKD compared to those on other regimens (HR = 2.4, 95% CI: 1.21–4.7) [53]. In contrast, NNRTIs were found to be 55% protective against CKD development compared to PI/r (OR = 0.45, 95% CI: 0.24–0.83) [41]. Prior antiretroviral therapy exposure (OR = 1.22, 95% CI: 1–1.5) [46], and longer duration of TDF exposure (OR = 26.3, 95% CI: 2.02–343.04 [86]; OR = 1.16, 95% CI: 1.04–1.3) [41] were positive predictor of CKD development. Additionally, a viral load of HIV RNA per 1log10 copies/ml higher was associated with almost four times the risk of developing CKD compared to their counterparts (OR = 4.41, 95% CI: 1.65–11.78) [31].

**Baseline renal function.** Baseline eGFR 60-89ml/min (HR = 1.7, 95% CI: 1.2–23) compared to eGFR >/ = 90ml/min [52] showed a significant association with CKD. Moreover, studies revealed significant associations between creatinine clearance per 10ml increase (OR = 0.8, 95% CI: 0.64–0.99), creatinine clearance for every 1ml decrease (RR = 0.95, 95% CI: 0.93–0.96) [51], hyperfiltration at baseline (HR = 4.1, 95% CI: 2.3–7.1) [52], higher serum creatine at baseline (OR = 49.8, 95% CI: 79–311.92) [41], and the presence of CKD.

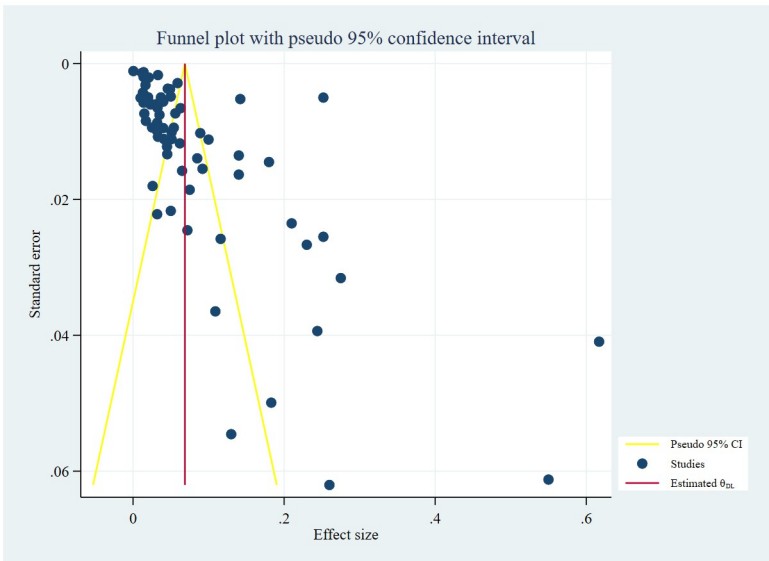

**Fig 7. Funnel plot.**

## Publication bias

Included studies showed publication bias demonstrated by a funnel plot (Fig 7). We confirmed the significance of publication bias using egger's statistical test (P <0.00001).

The contour enhanced funnel plot also showed publication bias (Fig 8).

To minimize the impact of publication bias on the pooled prevalence estimate of CKD, we performed a trim and fill analysis. We used a linear estimator with a fixed effect model to show imputed studies on left, followed by a random effects model to estimate the pooled prevalence of CKD using the trim and fill method. In this analysis, we found thirty-three imputed studies

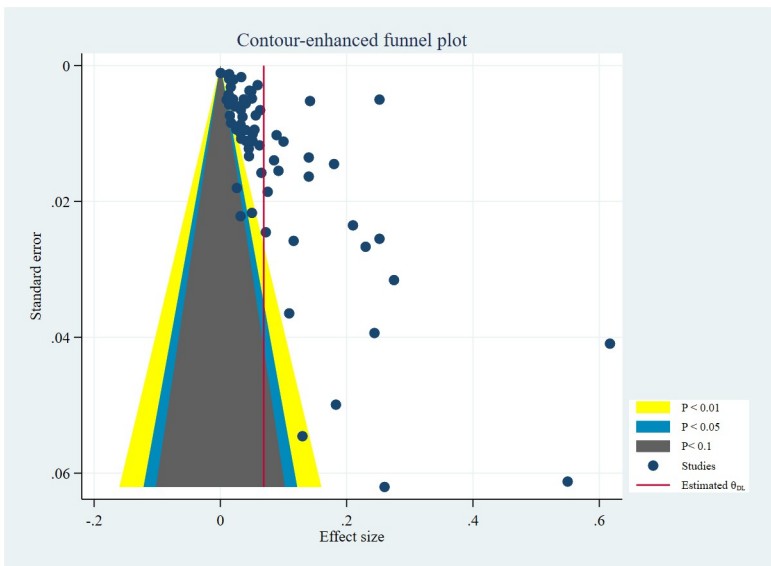

**Fig 8. Contour enhanced funnel plot.**

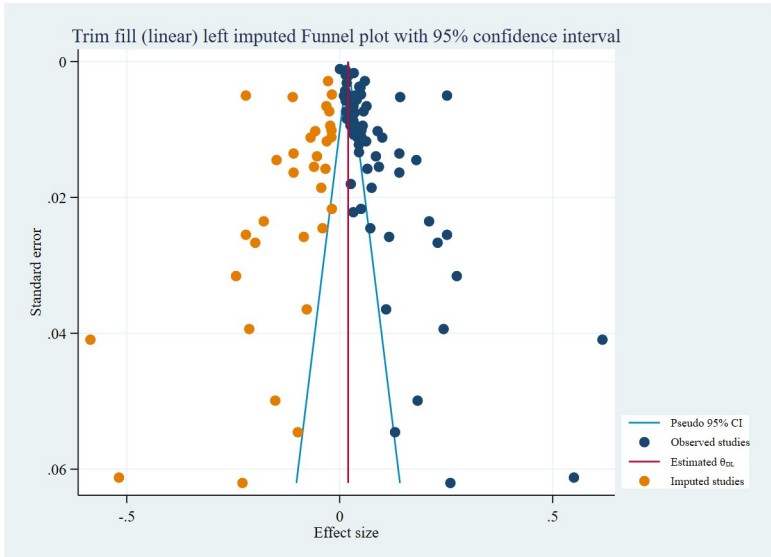

**Fig 9. Funnel plot using nonparametric trim-and-fill analysis of publication bias.**

on left, and 2% (95% CI: 1–3%) overall pooled prevalence of CKD using the trim and fill method (Fig 9).

## Risk of bias of included studies

Risk of bias was assessed for eligible studies based on Hoy et al., 2012 risk assessment tool and the results have shown in S4 Table.

## Discussion

We conducted the present review to address the data gap regarding the pooled prevalence of CKD among PLWHIV receiving TDF based regimens across the world. In this review, we found a 7% (95% CI: 6–8%) pooled prevalence of CKD. The current review revealed that the prevalence of CKD among PLWHIV varied both within and between regions. The highest prevalence (11.7% [95% CI:8.4–15]) was observed in Sub-Sahara Africa region, while it was lower (3% [95% CI:2–4.1]) in other regions (studies conducted in more than one region). Additionally, the highest prevalence (23% [95% CI: 18–28%]) was found in Tanzania, whereas China showed the lowest burden of pooled prevalence of CKD (1% [95% CI: 0–2%]).

The present review included studies that used different eGFR estimation methods, and CKD diagnostic criteria with respect to time. Subgroup analysis found that the pooled prevalence of CKD based on CG, MDRD, and MDRD without race factor was significantly different from the overall pooled prevalence of CKD. Included studies used different methods, such as on the spot CKD diagnosis without baseline data, or with normal baseline data, or a decline to <50 or 60ml/min at two time points separated by at least 3 months for CKD confirmation. Subgroup analysis found significant difference among these groups.

Moreover, we found that the pooled CKD prevalence in LMIC, studies conducted in more than one income level, prospective cohort studies, and clinical trials significantly differed from the overall prevalence of CKD. The variation within and between regions may be related to differences in baseline study characteristics (comorbidity, comedication, renal function, WHO

clinical stage of HIV/AIDS), duration of TDF exposure, eGFR estimation methods, income level, and study design [52, 56, 77, 86].

The interpretation of CKD in the present meta-analysis requires caution due to the high heterogeneity between studies. We performed sensitivity analysis by removing studies with potential outlier data, but heterogeneity and publication bias remained high, indicating unexplained heterogeneity.

The findings of this systematic review and meta-analysis are crucial for enhancing the quality of HIV care by providing a global picture of CKD in TDF based regimens, and suggesting strategies to prevent TDF related renal adverse effects. Furthermore, this review also has clinical importance in triggering healthcare policy makers worldwide to design strategies to optimize HIV care. Hence, optimizing TDF based regimens, and early CKD identification could help reduce further complication and mortality [93]. Our review finding suggests a re-evaluation of TDF based regimens in HIV care, which is in line with a systematic review summarizing nephrotoxicity in TDF based regimens across the globe [94]. Moreover, a scoping review supports our concern, highlighting a high burden of renal toxicity in TDF based regimens, and advocating for body weight or body mass index-based dosing [93]. A systematic review by Cooper RD et al. [10], and Mitsi TJ et al. [11], revealed a significant renal function decline in TDF based regimen, with a conclusion that such decline has modest clinical effect or was not enough to contradict its use. They recommend to consider consumer factors, and regular monitoring of renal function in TDF use [10, 11].

The mechanism of TDF induced nephrotoxicity is not well understood. However, the potential mechanism by which TDF causes kidney damage involves inhibition of mitochondrial DNA polymerase gamma [95]. Increased entry and decreased efflux of tenofovir by transport proteins in the renal tubule increase tenofovir induced renal toxicity [96].

Despite literature, and guidelines recommending the identification of patient factors, and regular monitoring of renal function in routine care, TDF initiation and use often occur without laboratory monitoring in low-income countries. In the present meta-analysis, we found significant association between being female, and CKD. This finding is in line with the evidences obtained from individual studies [50, 52, 56].

Moreover, CD4 count less than 200 copies/ml showed statistically significant association with CKD. These pooled effect supports the results from studies [50, 52, 56]. Studies of the present review showed factors associated with CKD, but we did not show their pooled effect due to either using different predictor category or statistical predictor models. These studies revealed that WHO stage [51, 86], cancer [77], HIV RNA viral load [31], blood pressure [19], SCr [41], eGFR [52], diabetes mellitus [56], ritonavir boosted protease inhibitor use [53], prior ART exposure [46], and TDF exposure duration [41, 86] have significant association with CKD in TDF based regimen. Health care providers should be vigilant to assess the risk of renal impairment in PLWHIV taking TDF. This review recommends monitoring of renal function before and during TDF use. In addition, it warrants the consideration gender at birth, and low CD4 count in routine health care practice to prevent CKD. Tenofovir alafenamide (TAF) is another prodrug formulation of tenofovir has better renal safety (but still has a concern) and equivalent efficacy compared to TDF [97, 98]. This safety difference was reported in reviews where TAF, and TDF have unique safety when they are used with pharmacokinetics boosters, but such difference was not seen when unboosted TAF compared to unboosted TDF [97, 98]. Evidence showed that using TDF has beneficial effect regarding lipid profiles [70], whereas TAF increases them [98]. Still guidelines consider both formulations as components of preferred regimens. High level evidences are demanded regarding overall safety, cost, and access of TDF versus TAF to modify clinical guidelines across the globe [98]. We recommend a large-

scale post market study of TDF based regimen that help health policymakers to provide evidence-based decision.

This finding indicates high public health burden of CKD. Our finding has clinical implication that safety of TDF in HIV care where there is no regular baseline, and following up renal function monitoring is questionable, which is supported by other literature [94]. Considering this, and its high resource demanding management: we recommend to have a consensus on eGFR estimation equation to assess CKD in research, and health care practice as our review showed using different equations result in different results. The safety of TDF should be re-evaluated with high level evidences. In addition, health life style practice along with regular renal function monitoring has to be integrated with routine HIV care to prevent CKD.

Our review is the first to show the pooled estimate of CKD prevalence worldwide. However, it has limitations such as the CKD definition was not considering proteinuria or albuminuria; it was defined using only eGFR. Studies that estimated CKD on the spot were included, which may lead to under or over estimation of the pooled effect. In addition, studies did not give TDF with NNRTI, and PI/r specific data, so we did not show the pooled estimate of CKD in such subcategory. The included studies have high heterogeneity, and publication bias. we only included studies written in English. Further, we did non parametric trim and fill method analysis to minimize the impact of publication bias on the overall prevalence of CKD. Following trim and fill analysis, we found the pooled prevalence of CKD corrected for publication bias, which was 2% (95% CI: 1–3%).

## Conclusion

The present systematic review found a considerably high prevalence of CKD among HIV patients receiving TDF based regimens. A CD4 count of less than 200 copies per ml, and being female were significant predictor of CKD. Thus, we recommend regular renal function monitoring for PLWHIV receiving TDF. especially those with low CD4 counts, and females, to prevent, identify and manage CKD.

## Supporting information

**S1 Table. PRISMA 2020 checklist.**
(DOCX)

**S2 Table. Search strategy.**
(DOCX)

**S3 Table. Methodological quality assessment score of included studies.**
(DOCX)

**S4 Table. The risk of bias assessment tool results for the included studies.**
(DOCX)

**S5 Table. All accessed studies in literature search.**
(XLSX)

**S6 Table. Data extracted from included studies.**
(XLSX)

## Acknowledgments

For the present systematic review, we prospectively registered a systematic review protocol in PROSPERO with a registration number of CRD42020136813. It should be pointed out that the

systematic review at hand has not involved any direct human or animal subjects and has rather synthesized secondary data. All data sources were public and referenced. The authors, therefore, did not need ethical approval.

## Author Contributions

**Conceptualization:** Taklo Simeneh Yazie, Wondimeneh Shibabaw Shiferaw, Abebe Muche Belete.

**Data curation:** Taklo Simeneh Yazie, Wondimeneh Shibabaw Shiferaw, Asaye Alamneh Gebeyehu, Assefa Agegnehu Teshome, Zenaw Debasu Addisu, Abebe Muche Belete.

**Formal analysis:** Wondimeneh Shibabaw Shiferaw, Asaye Alamneh Gebeyehu, Assefa Agegnehu Teshome, Zenaw Debasu Addisu, Abebe Muche Belete.

**Funding acquisition:** Taklo Simeneh Yazie, Abebe Muche Belete.

**Investigation:** Taklo Simeneh Yazie, Wondimeneh Shibabaw Shiferaw, Asaye Alamneh Gebeyehu, Assefa Agegnehu Teshome, Zenaw Debasu Addisu, Abebe Muche Belete.

**Methodology:** Taklo Simeneh Yazie, Wondimeneh Shibabaw Shiferaw, Asaye Alamneh Gebeyehu, Assefa Agegnehu Teshome, Zenaw Debasu Addisu, Abebe Muche Belete.

**Project administration:** Taklo Simeneh Yazie.

**Resources:** Wondimeneh Shibabaw Shiferaw, Abebe Muche Belete.

**Supervision:** Abebe Muche Belete.

**Validation:** Taklo Simeneh Yazie, Wondimeneh Shibabaw Shiferaw, Asaye Alamneh Gebeyehu, Assefa Agegnehu Teshome, Zenaw Debasu Addisu, Abebe Muche Belete.

**Visualization:** Taklo Simeneh Yazie, Wondimeneh Shibabaw Shiferaw, Asaye Alamneh Gebeyehu, Assefa Agegnehu Teshome, Zenaw Debasu Addisu, Abebe Muche Belete.

**Writing – original draft:** Taklo Simeneh Yazie, Wondimeneh Shibabaw Shiferaw, Assefa Agegnehu Teshome, Zenaw Debasu Addisu, Abebe Muche Belete.

**Writing – review & editing:** Taklo Simeneh Yazie, Wondimeneh Shibabaw Shiferaw, Asaye Alamneh Gebeyehu, Assefa Agegnehu Teshome, Zenaw Debasu Addisu, Abebe Muche Belete.

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
