## [Decision Letter · Decision Letter 0]

14 Oct 2024

PONE-D-24-20050Chronic kidney disease among people living with HIV on TDF based regimen: A Systematic review and meta-analysisPLOS ONE

Dear Dr. Yazie

Thank you for submitting your manuscript to PLOS ONE. After careful consideration, we feel that it has merit but does not fully meet PLOS ONE’s publication criteria as it currently stands. Therefore, we invite you to submit a revised version of the manuscript that addresses the points raised during the review process.

**ACADEMIC EDITOR: **Please review the syntax and the language of the manuscript

We look forward to receiving your revised manuscript.

Kind regards,

Tarek Samy Abdelaziz, MD

Academic Editor

PLOS ONE

Journal Requirements:

3. As required by our policy on Data Availability, please ensure your manuscript or supplementary information includes the following:

Reviewers' comments:

Reviewer's Responses to Questions

**Comments to the Author**

1. Is the manuscript technically sound, and do the data support the conclusions?

Reviewer #1: Yes

2. Has the statistical analysis been performed appropriately and rigorously? 

Reviewer #1: Yes

3. Have the authors made all data underlying the findings in their manuscript fully available?

Reviewer #1: Yes

4. Is the manuscript presented in an intelligible fashion and written in standard English?

Reviewer #1: Yes

5. Review Comments to the Author

Reviewer #1: The manuscript by Yazie et al. “Chronic kidney disease among people living with HIV on TDF based regimen: A Systematic review and meta-analysis” looks interesting. This systematic review article aims to determine the prevalence of CKD among PLWHIV receiving a TDF based regimen using publications between January 2000 and May 2024. The meta-analysis in this article included 69 studies with 88299 participants and found that the pooled prevalence of CKD was 7% (95% CI:6-8) whereas CD4 count was less than 200 copies per milliliter in females associated with CKD. The authors concluded that the magnitude of CKD across the globe is high in people with HIV and who have received TDF based regimen.

The manuscript is well written and provides enough strategies to fulfill its aims, though language may need some polishing for better flow. In conclusion, I support publication of this systematic review and believe that it would be interesting for researchers working in the field.

6. PLOS authors have the option to publish the peer review history of their article (what does this mean?). If published, this will include your full peer review and any attached files.

Reviewer #1: No

---

## [Author Response · Author response to Decision Letter 0]

1 Dec 2024

To The Editor, Reviewers

PLOS ONE

Date: Nov 26, 2024

Subject: Addressing Editor/Reviewer Comments for PONE-D-24-20050

Dear Editor, Reviewers,

We would like forwarding our great thanks for your crucial feedback to improve the status of the paper. We took long time to address the comments provided. Below, we are providing responses and actions to address each comment in a step-by-step way and in detail of our best ability. The responses are highlighted in blue.

Kind regards,

Authors of PONE-D-24-20050

PONE-D-24-20050

Chronic kidney disease among people living with HIV on TDF based regimen: A Systematic review and meta-analysis

PLOS ONE

Dear Dr. Yazie

Thank you for submitting your manuscript to PLOS ONE. After careful consideration, we feel that it has merit but does not fully meet PLOS ONE’s publication criteria as it currently stands. Therefore, we invite you to submit a revised version of the manuscript that addresses the points raised during the review process.

ACADEMIC EDITOR:

Please review the syntax and the language of the manuscript

o Included

o Included 

o Included 

We look forward to receiving your revised manuscript.

Kind regards,

Tarek Samy Abdelaziz, MD

Academic Editor

PLOS ONE

Journal Requirements:

o We have checked that the manuscript meets the PLOS ONE journal requirements.

o We have included captions for all supporting information.

3. As required by our policy on Data Availability, please ensure your manuscript or supplementary information includes the following:

o We have included a numbered table in excel of all studies identified in literature search (S5 Table). Reasons of exclusion have been included in this table of excel.

o We have included reasons of exclusion for all excluded studies that are identified in literature search (S5 Table).

o This is not applicable in this review, as no unpublished studies were identified during the literature search.

o We have included it as per suggestion (S6 Table).

o We attempted to contact the corresponding authors obtain missing data for some studies, but authors did not respond to our requests. Therefore, in this review, there are no data obtained from another sources.

o We have included the methodological quality assessment results for randomized clinical trials based on Cochrane risk-of-bias tool for randomized trials. Each domain has been assessed and also answers were provided for each signalling question within each domain (included under S3 File).

o In this review, we used a complete-case analysis approach to include only studies that had complete data for the outcome of interest. For studies with missing data, we tried to contact the corresponding authors to obtain the missing information. However, despite these efforts, none of them responded to our requests.

o We have checked all references for their completeness and correctness. No retracted articles found in the included references.

Reviewers' comments:

Reviewer's Responses to Questions

Comments to the Author

1. Is the manuscript technically sound, and do the data support the conclusions?

Reviewer #1: Yes

2. Has the statistical analysis been performed appropriately and rigorously?

Reviewer #1: Yes

3. Have the authors made all data underlying the findings in their manuscript fully available?

Reviewer #1: Yes

4. Is the manuscript presented in an intelligible fashion and written in standard English?

Reviewer #1: Yes

5. Review Comments to the Author

Reviewer #1: The manuscript by Yazie et al. “Chronic kidney disease among people living with HIV on TDF based regimen: A Systematic review and meta-analysis” looks interesting. This systematic review article aims to determine the prevalence of CKD among PLWHIV receiving a TDF based regimen using publications between January 2000 and May 2024. The meta-analysis in this article included 69 studies with 88299 participants and found that the pooled prevalence of CKD was 7% (95% CI:6-8) whereas CD4 count was less than 200 copies per milliliter in females associated with CKD. The authors concluded that the magnitude of CKD across the globe is high in people with HIV and who have received TDF based regimen.

The manuscript is well written and provides enough strategies to fulfill its aims, though language may need some polishing for better flow. In conclusion, I support publication of this systematic review and believe that it would be interesting for researchers working in the field.

o Dear Reviewer, we thank you for your encouragement and helpful comments. We have done our best to improve the language of this paper.

6. PLOS authors have the option to publish the peer review history of their article (what does this mean?). If published, this will include your full peer review and any attached files.

Do you want your identity to be public for this peer review? For information about this choice, including consent withdrawal, please see our Privacy Policy.

Reviewer #1: No

---

## [Editor Report · Decision Letter 1]

10 Jan 2025

Chronic kidney disease among people living with HIV on TDF based regimen: A Systematic review and meta-analysis

PONE-D-24-20050R1

Dear Dr. Yazie,

We’re pleased to inform you that your manuscript has been judged scientifically suitable for publication and will be formally accepted for publication once it meets all outstanding technical requirements.

Kind regards,

Tarek Samy Abdelaziz, MD

Academic Editor

PLOS ONE
---

## [Editor Report · Acceptance letter]

24 Jan 2025

PONE-D-24-20050R1 

PLOS ONE

Dear Dr. Yazie, 

I'm pleased to inform you that your manuscript has been deemed suitable for publication in PLOS ONE. Congratulations! Your manuscript is now being handed over to our production team.

Kind regards, 

on behalf of

Professor Tarek Samy Abdelaziz 

Academic Editor

PLOS ONE